# Specific pelvic shape in patients with developmental dysplasia of the hip on 3D morphometric homologous model analysis

Yui Sasaki[1], Daisuke Suzuki[2]*, Ryo Tokita[3], Hiroyuki Takashima[4], Hirofumi Matsumura[5], Satoshi Nagoya[6]

1 Division of Rehabilitation, Hitsujigaoka Hospital, Sapporo, Japan, 2 Department of Health Sciences, Hokkaido Chitose Collage of Rehabilitation, Chitose, Japan, 3 Department of Rehabilitation, Sapporo Medical University Hospital, Sapporo, Japan, 4 Division of Biomedical Science and Engineering, Faculty of Health Sciences, Hokkaido University, Sapporo, Japan, 5 Department of Physical Anthropology, School of Health Science, Sapporo Medical University, Sapporo, Japan, 6 Division of Orthopaedic Surgery, Sapporo Kojinkai Memorial Hospital, Sapporo, Japan

* d-suzuki@chitose-reha.ac.jp

## Abstract

### Purpose

To clarify the morphological factors of the pelvis in patients with developmental dysplasia of the hip (DDH), three-dimensional (3D) pelvic morphology was analyzed using a template-fitting technique.

### Methods

Three-dimensional pelvic data of 50 patients with DDH (DDH group) and 3D pelvic data of 50 patients without obvious pelvic deformity (Normal group) were used. All patients were female. A template model was created by averaging the normal pelvises into a symmetrical and isotropic mesh. Next, 100 homologous models were generated by fitting the pelvic data of each group of patients to the template model. Principal component analysis was performed on the coordinates of each vertex (15,235 vertices) of the pelvic homologous model. In addition, a receiver-operating characteristic (ROC) curve was calculated from the sensitivity of DDH positivity for each principal component, and principal components for which the area under the curve was significantly large were extracted (p<0.05). Finally, which components of the pelvic morphology frequently seen in DDH patients are related to these extracted principal components was evaluated.

### Results

The first, third, and sixth principal components showed significantly larger areas under the ROC curves. The morphology indicated by the first principal component was associated with a decrease in coxal inclination in both the coronal and horizontal planes. The third principal component was related to the sacral inclination in the sagittal plane. The sixth principal component was associated with narrowing of the superior part of the pelvis.

**Data Availability Statement:** All relevant data are within the manuscript and its Supporting Information files.

**Funding:** Grant-in-Aid for Scientific Research(C) from Japan Society for the Promotion of Science. Grant Number: 21K11269. The funders had no role in study design, data collection and analysis, decision to publish, or preparation of the manuscript.

**Competing interests:** The authors have declared that no competing interests exist.

## Conclusion

The most important factor in the difference between normal and DDH pelvises was the change in the coxal angle in both the coronal and horizontal planes. That is, in the anterior and superior views, the normal pelvis is a triangle, whereas in DDH, it was more like a quadrilateral.

## Introduction

Developmental dysplasia of the hip (DDH) refers to a condition in which the acetabulum that covers the femoral head is hypoplastic, and it is defined as a center-edge (CE) angle of less than 20 degrees [1]. DDH is more common in women and is considered a cause of hip osteoarthritis, which has a prevalence of approximately 3.5% in Japanese women [2]. In fact, approximately 80% of Japanese women with osteoarthritis of the hip have DDH, and many of them develop the osteoarthritis at an early age [3]. DDH also causes pain and instability in the hip joint, which can be problematic in daily life. Many patients with DDH also have low back pain [4–7].

With the development of diagnostic imaging technology, the location of acetabular defects in patients with DDH is not uniform and can be divided into three types: total defect, anterosuperior defect, and posterosuperior defect [8, 9]. Approximately 18% have been shown to have insufficient posterior and posterosuperior coverage of the acetabulum [10]. Furthermore, the acetabular notch in patients with posterosuperior defects has been shown to be located more anteriorly than in healthy subjects [11]. In other words, the pelvis of DDH patients does not have a fixed area of dysplasia, but the location of the dysplasia differs depending on the patient. Furthermore, DDH is known to cause not only dysplasia of the acetabulum, but also deformity of the entire pelvis. For example, it has been reported that the pelvis is tilted more forward than in healthy individuals, both superior and inferior iliac wing angles are larger [12], the width of the pelvic outlet is wider [13], and the iliac wings are curved inwardly [14]. However, at present, evaluation of pelvic morphology in DDH patients is limited to local measurements, and morphological analysis of the entire pelvis has not been performed.

In recent years, advances in computer processing power and algorithms have made it possible to analyze entire 3D image data. Of such analysis methods, template fitting is attracting attention [15]. In this study, template fitting was used to perform a homologous model analysis of the pelvis. The homologous model analysis is a method that comprehensively measures how much and in what direction the model changes. This made it possible to extract the characteristic shape of the pelvis of DDH patients.

## Methods

### Subjects

Three-dimensional pelvic data of 50 patients with DDH (DDH group) and 3D pelvic data of 50 patients without obvious pelvic deformity (Normal group) were used with computed tomography (CT Aquilion CX Edition, Toshiba Medical Systems, Otawara, Japan; tube voltage 120 kV; slice thickness 0.5 mm). The inclusion criteria were female patients who visited Sapporo Medical University Hospital between April 2013 and December 2021 for both the DDH and Normal groups, had no indication of opt-out, had no history of hip or spine surgery, and were under 55 years of age. The DDH group consisted of patients who underwent rotational acetabular osteotomy (RAO) at Sapporo Medical University. They were all Tönnis grade 0 or

1. The Normal group consisted of patients with no obvious pelvic deformity and a center-edge angle of 25 degrees or more. The mean age was 39.4 years (range 21–51 years) in the DDH group and 39.0 years (range 21–52 years) in the Normal group.

In conducting this study, the normal group were used anonymized CT DICOM data. The DDH group were used CT DICOM data anonymized to prevent identification by authors other than the attending physician (SN). These CT DICOM data were accessed after approval by the IRB on February 10, 2022. We used the data stored by the Department of Orthopedic surgery, Sapporo Medical University. Since we used patient data obtained during regular medical treatment, the Ethics Committee of our institution has approved that an opt-out format is acceptable without requiring informed consent from individual patients. This study was approved by the institutional review board of Sapporo Medical University Hospital (approval number: 322–205).

## Template fitting

Although a 3D model consists of a polygon mesh, there are basically no homologous vertices between a template model and an arbitrary 3D model, and the number of vertices is not identical, so it is impossible to compare them as they are. Therefore, homologous modeling by template fitting was used to make arbitrary 3D models comparable.

In template fitting, first, homologous points such as protrusions, and ridges, etc. are registered as landmarks in both the reference model (template model) and the 3D models to be analyzed. Next, a homologous model is created by transforming and superimposing the template model into the 3D model to be analyzed, based on the landmarks as a reference. The created homologous model can be assumed to show overall homology with the template model (Fig 1). Therefore, by creating homologous models of all the pelvises to be analyzed, it is possible to comprehensively evaluate in which direction and degree each vertex changes among the homologous modeled pelvises.

The 3D pelvis model was reconstructed using 3D imaging software (Mimics ver. 23.0, Materialize, Leuven, Belgium) from a series of CT images and output by converting to STL format. Since a shape with holes cannot create a homologous model, the anterior and posterior sacral foramina and vertebral foramina were closed using 3D modeling software (3matic ver. 15.0, Materialize). After these preparations, 10 randomly selected pelvises were synthesized, mirrored to form them bilaterally symmetrically, and isotropically remeshed to create the template model. This resulted in the creation of a template model with 15,235 vertices. This template model was placed in the anterior pelvic plane [16, 17]. Next, 58 points consisting of the anterior superior iliac spine and symphysis pubis, etc. were defined as landmarks (Fig 2).

The pelvises to be analyzed were placed in the anterior pelvic coordinate system after filling the holes in the same way as the template model. Next, template fitting was performed using the landmarks as a reference to create a homologous model. The homologous model created using this template fitting had the same topology and number of data points as the template model (Fig 2).

The homologous model was created using HBM-Rugle software (Medic Engineering, Kyoto, Japan, http://www.rugle.co.jp/). The main parts of this software were originally developed at the Digital Human Research Center of the National Institute of Advanced Industrial Science and Technology [18]. The robustness of the fitting accuracy of this software has been verified in several studies [19–26].

In the actual template fitting using HBM-Rugle, the pelvis model to be analyzed was superimposed on the template model using the ICP method [27] with 58 landmarks as reference points. Next, the template model was fitted to the pelvis model to be analyzed using non-rigid

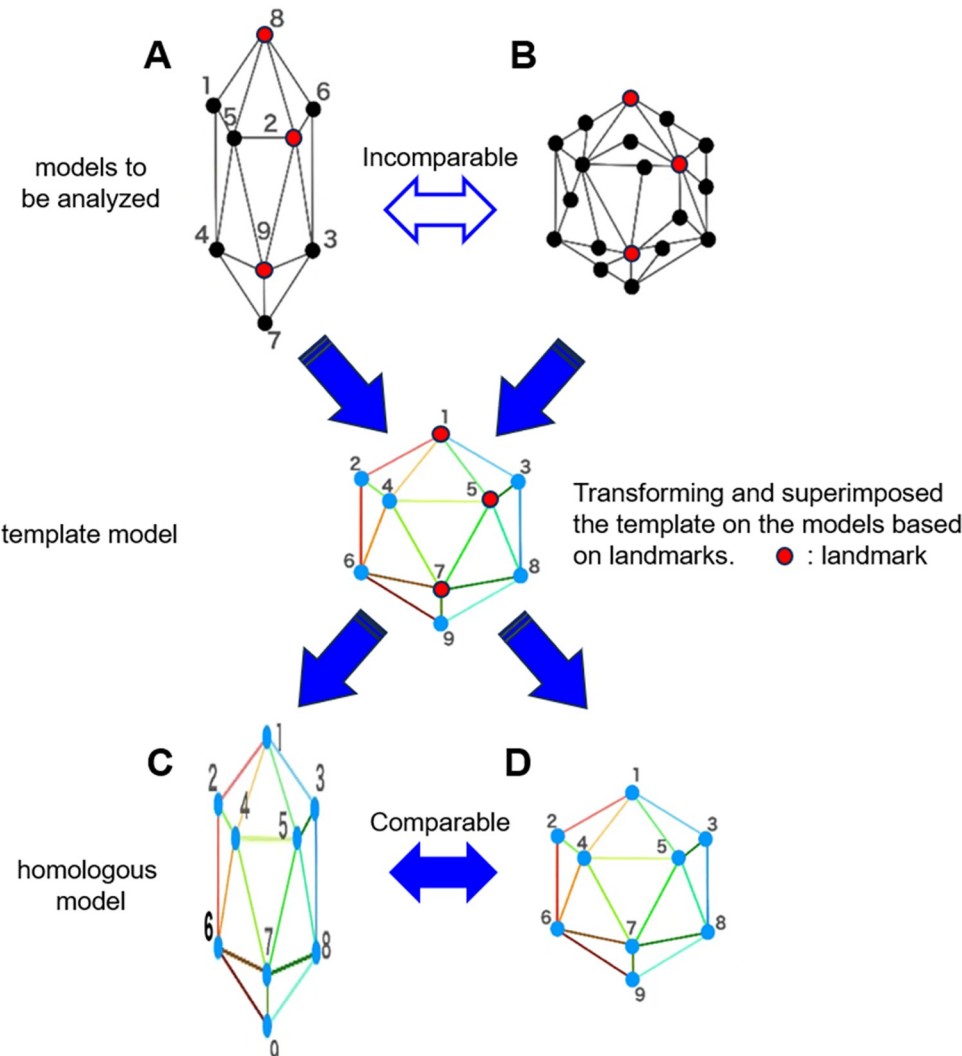

**Fig 1. Creation of a homologous model by template fitting.** If the models to be analyzed (A, B) have different shapes and numbers of vertices, they cannot be compared. Therefore, the template model is transformed and superimposed based on landmarks into the models to be analyzed. The created model is called a homologous model, and it can be assumed that all vertices are homologous to the template model. Comparisons can be made by converting all models to be analyzed into homologous models (C, D).

mesh deformation to generate a homologous model. The average distance between the planes of the generated homologous model and the original pelvis model was 0.632±0.124 mm.

## Data analyses

The size of all homologous pelvis models was normalized to minimize the sum of squared distances between each vertex of the template model and each vertex of the homologous model. Principal component analysis was then performed on the three-dimensional coordinates of the vertices that made up each model (HBM-Rugle).

Of the principal components obtained, receiver-operating characteristic (ROC) curve analysis was performed to evaluate the characteristics found in pelvises with DDH [28]. The ROC curves were determined by setting the DDH group to 1 and the Normal group to 0 for the

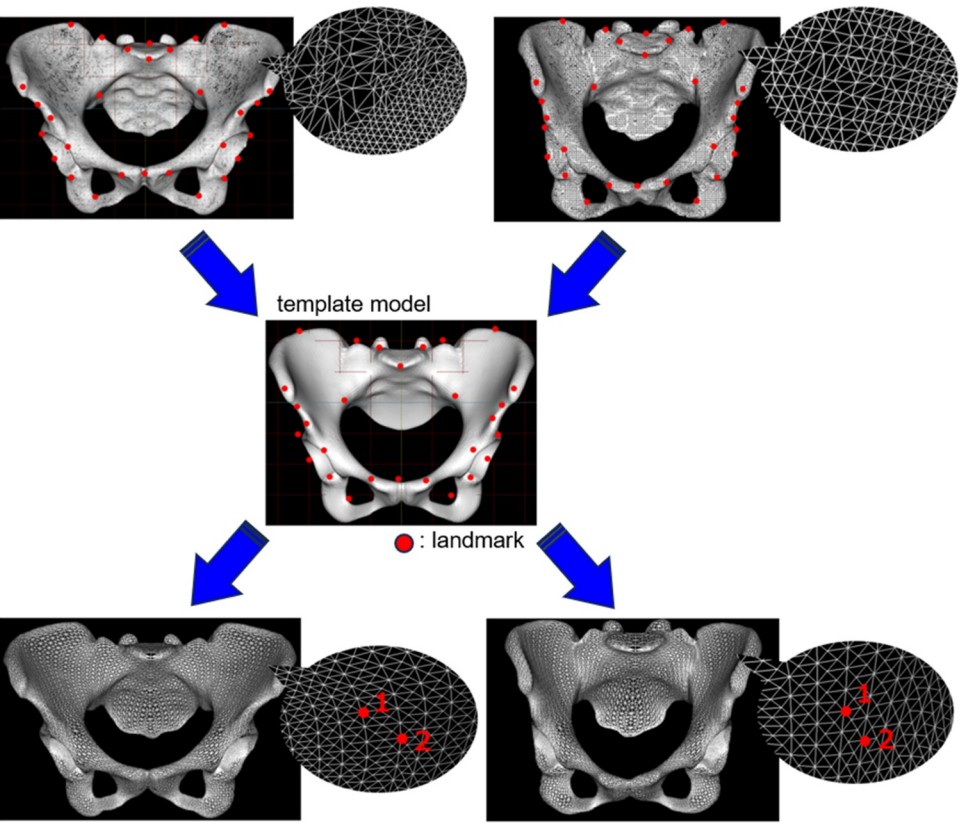

**Fig 2. Homologous modeling of the actual pelvis.** A homologous model with corresponding vertices is created from a template consisting of 15,235 vertices.

principal component score of each pelvic model. The area under the curve (AUC) of the ROC curve was determined, and then the significance of the AUC was evaluated using the $\chi^2$ test (null hypothesis = 0.5). The significance level was set at 5%. The ROC analysis was performed using Bell Curve for Excel (version 3.21, SSRI, Tokyo, Japan). Next, virtual pelvic morphology that would occur when the principal component score was changed by ±3 standard deviations (SD) was outputted, and how each principal component affected the pelvic morphology was evaluated (HBM-Rugle). In addition, to evaluate the meaning of each principal component, a scatter plot of each measurement value of the pelvis (sharp angle, iliac wing angle, sacral slope, ischiopubic angle, pelvic inclination) when the pelvis is placed in the functional pelvic coordinate system was created [29] (Fig 3).

## Results

Principal component analysis was performed using the 3D coordinates of the vertices of the homologous pelvic model. When calculating up to the 20th principal component, the cumulative contribution rate exceeded 80%. ROC curve analysis was performed on these principal components with the DDH group as positive. The principal components with significant AUCs were the first (p<0.001), third (p<0.05), and sixth principal components (p<0.01) (Table 1 and Fig 4). Fig 5 shows histograms of the principal component scores (PC1, PC3, and PC6) that were significantly different, separately for the Normal group and the DDH group.

The virtual morphologies when these principal component scores were changed by ±3SD are shown in Figs 6–8. The morphology indicated by the first principal component was

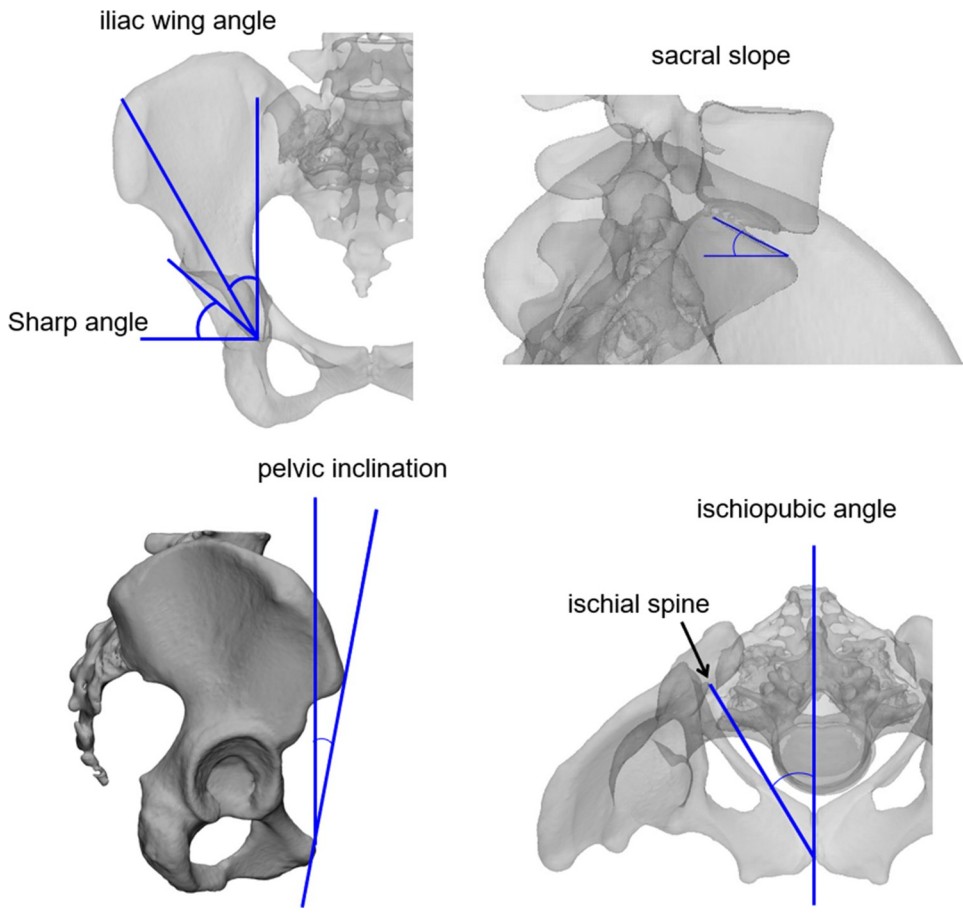

**Fig 3. The pelvic angles measured in this study.**

associated with a decrease in coxal inclination in both coronal and horizontal planes. In addition, it was associated with both an increased Sharp angle and an increased ischiopubic angle. The third principal component was related to posterior translation of the sacrum in the sagittal plane and a decrease in the sacral slope. The sixth principal component was associated with narrowing of the superior part of the pelvis.

Scatter plots between the PCs and the pelvic measurements were created, and it was found that PC1 had some strong correlations with each measurement except for the sacral slope, but

**Table 1. List of principal components up to the 6th order.**

| principal component | contributing ratio (%) | accumulative contributing ratio (%) | AUC | p value of $\chi^2$ test (null hypothesis: AUC = 0.5) |
|---|---|---|---|---|
| 1 | 16.785 | 16.785 | 0.8692 | < 0.001** |
| 2 | 12.497 | 29.282 | 0.5904 | 0.1170 |
| 3 | 9.471 | 38.753 | 0.6152 | 0.0418* |
| 4 | 5.485 | 44.239 | 0.5284 | 0.6265 |
| 5 | 4.356 | 48.595 | 0.5440 | 0.4531 |
| 6 | 4.142 | 52.737 | 0.6532 | 0.0058** |

* indicates p<0.05

** indicates p<0.01

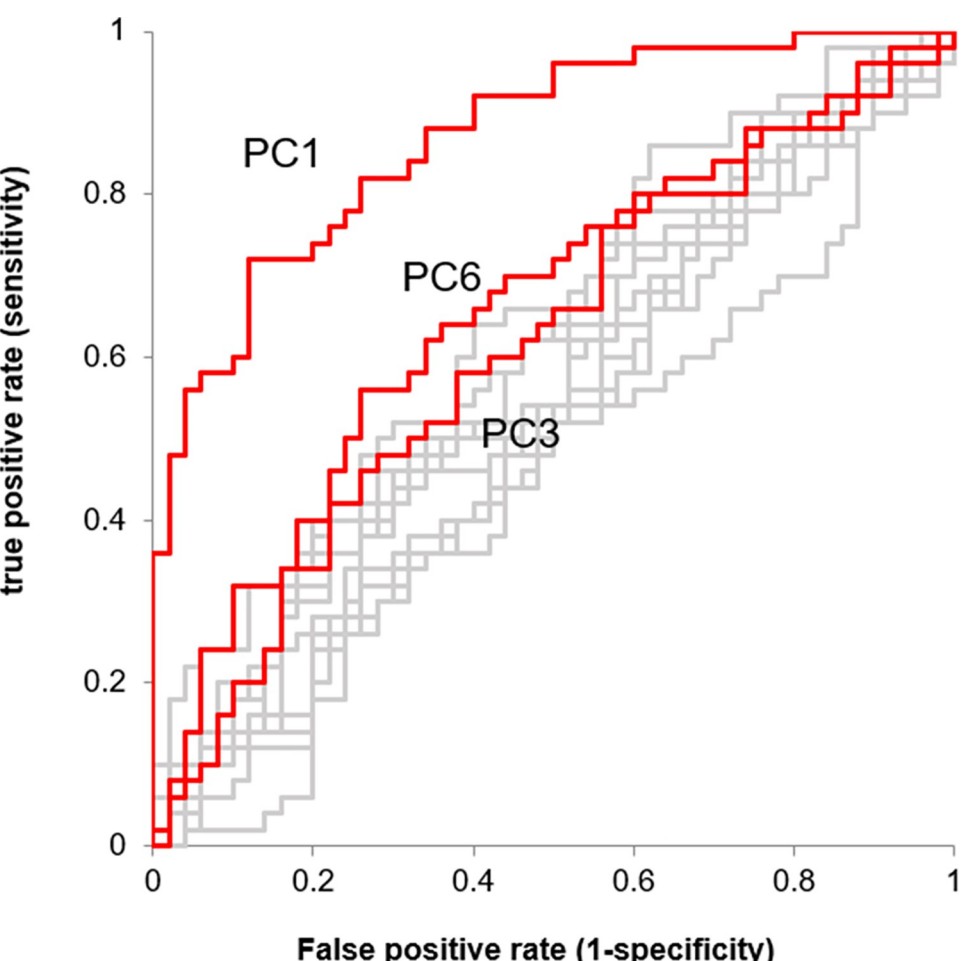

**Fig 4. The ROC curve of DDH for 1st-20th principal components.** The 1st, 3rd, and 6th principal components are shown in red.

strong correlations were not obtained for PC3 and PC6. Nevertheless, PC3 had a weak correlation with the Sharp angle, and PC6 had weak correlations with the iliac wing angle, sacral slope, and ischiopubic angle (Fig 9 and Table 2).

## Discussion

In this study, 100 homologous models were created by template fitting from both 50 DDH patients and 50 normal control patients. Morphological analysis for these homologous models was performed to extract the characteristics of the pelvis of DDH patients. The pelvic morphology of DDH patients was found to be different from the Normal group in terms of not only acetabular dysplasia, but also the inclination of the coxa and the sacrum. Although morphological analysis using a homologous model assumes that all vertices comprising the pelvis are homologous, it is an objective method that uses all morphological information. This method has recently been used to evaluate foot form, cranial morphology, facial aging changes, and pelvis sex determination, and it is attracting attention as a new comparison method for 3D models [15, 19, 21, 24].

DDH is a frequent hip disorder in Japan, and it has been reported that 80% of patients diagnosed with osteoarthritis of the hip joint have DDH [3]. It is also known that DDH occurs

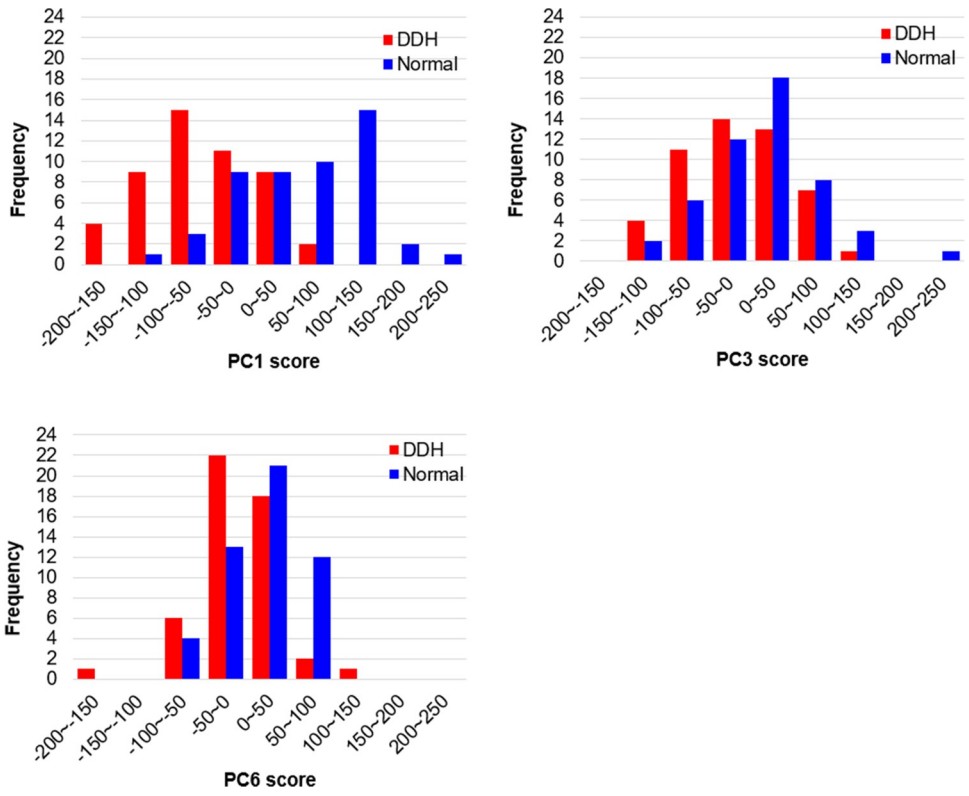

**Fig 5. Histograms of principal component scores (PC1, PC3, and PC6).**

more frequently in women. DDH is a multifactorial disease, and in addition to genetic factors, it has been suggested that the pressure on the hip joint applied during pregnancy, such as breech position, oligohydramnios, and overweight, is significant [30–32].

Although some differences in the morphology of the pelvis between healthy subjects and patients with DDH have been previously reported, comprehensive changes in the pelvic

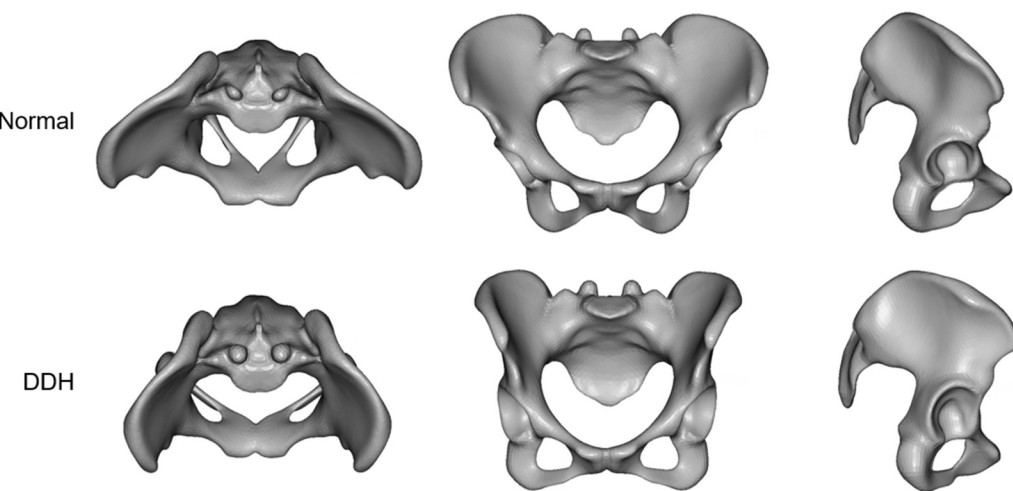

**Fig 6. The virtual morphologies when the first principal component (PC1) is changed from -3SD to +3SD.**

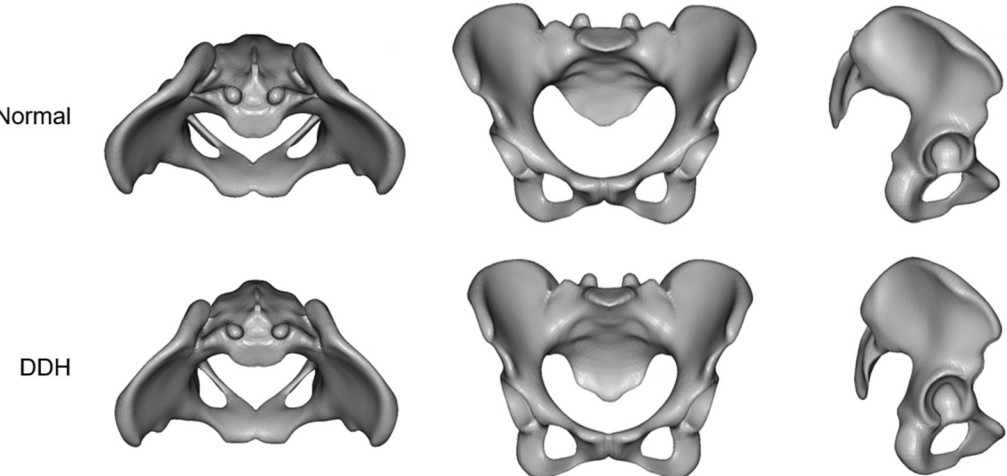

**Fig 7. The virtual morphologies when PC3 is changed from -3SD to +3SD.**

morphology have not been studied [12]. In the present study, it was possible to evaluate the characteristics of DDH in the overall morphology of the pelvis by creating a homologous model of the pelvis. On principal component analysis and ROC analysis using three-dimensional coordinates as an element, the AUCs of the ROC curves based on each principal component score of the first, third, and sixth principal components were significant.

The first principal component was associated with a decrease in inclination of the coxa in both coronal and horizontal planes (Fig 6). Although these have been reported previously [10, 14], they were evaluated separately, such as changes in the angle of the iliac wings and increases in the ischiopubic angle (Fig 9) [12]. However, the present results suggest that these differences stem from a single factor. The scatter plots showed that the first principal component also had a strong negative correlation with the Sharp angle (Fig 9) [33]. That is, as the coronal inclination of the coxa decreases, the acetabulum inclines, and in addition, coverage of the femoral head decreases.

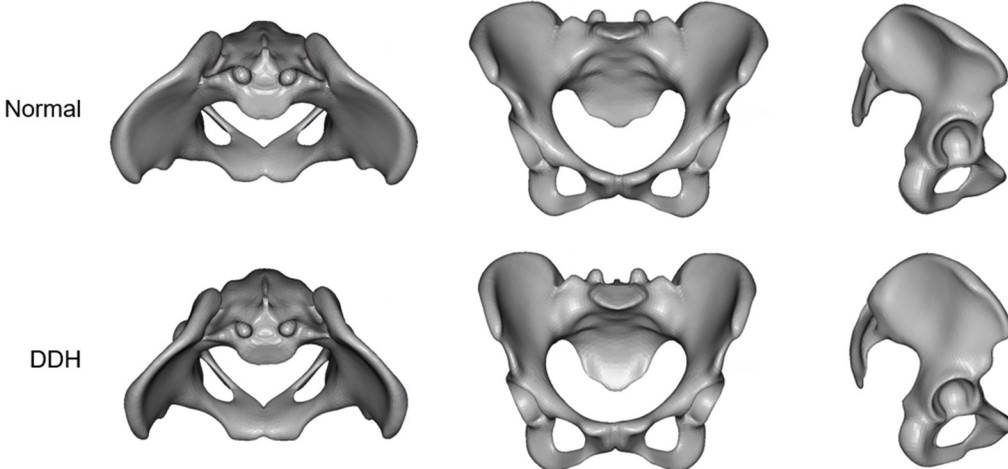

**Fig 8. The virtual morphologies when PC6 is changed from -3SD to +3SD.**

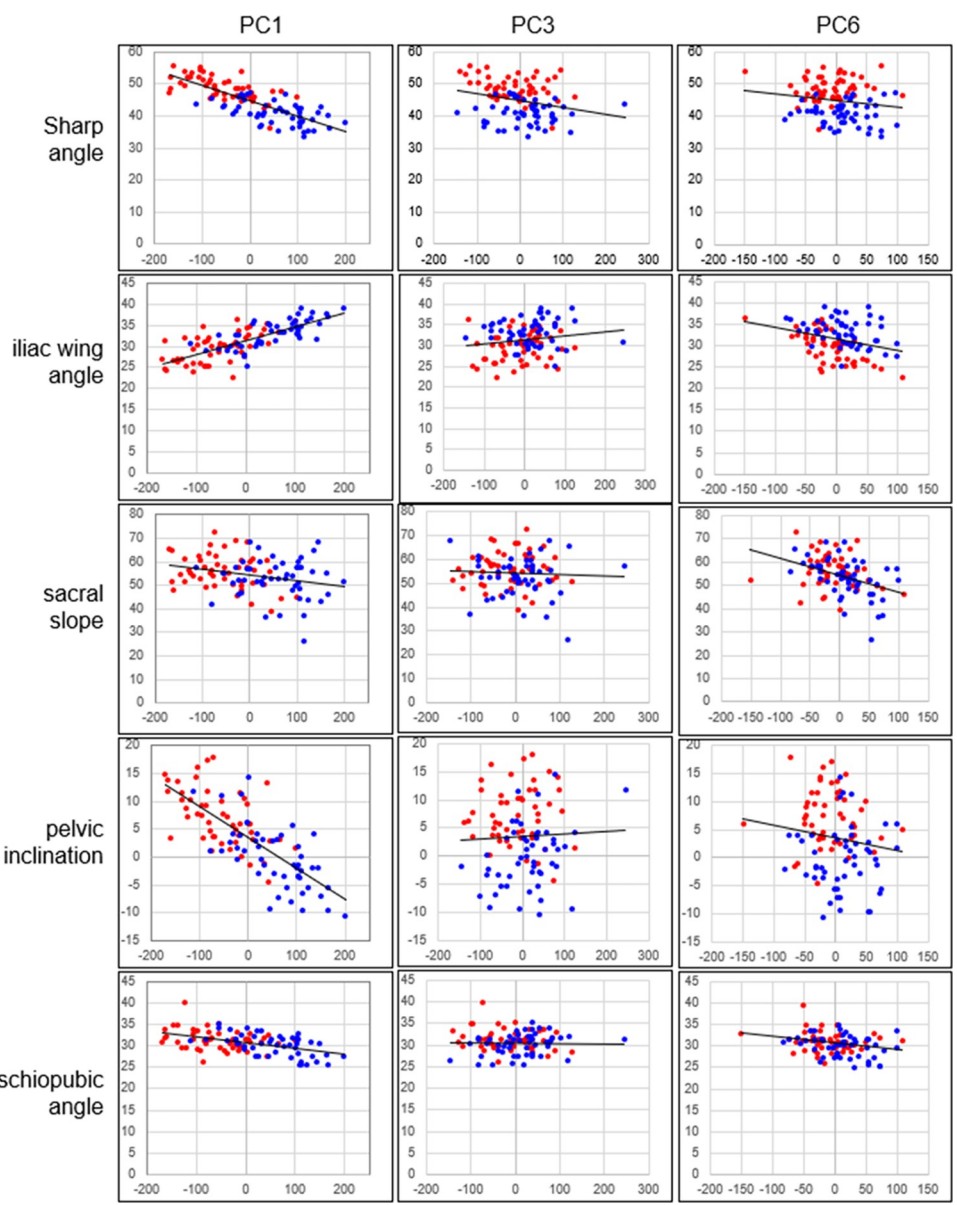

**Fig 9. Scatter plots between the PC1, PC3, and PC6 and the pelvic measurements.** Blue dots: the Normal group, Red dots: the DDH group.

**Table 2. Correlation coefficients between the PCs and the pelvic measurements.**

| Measurements | PC1 | PC3 | PC6 |
|---|---|---|---|
| Sharp angle | 0.7795 | 0.2619 | 0.1685 |
| Iliac wing angle | 0.7463 | 0.1783 | 0.3095 |
| Sacral slope | 0.2711 | 0.0616 | 0.3959 |
| Pelvic inclination | 0.7469 | 0.0424 | 0.1520 |
| Ischiopubic angle | 0.4802 | 0.0045 | 0.2735 |

The third principal component appeared to be related to sacral slope (Fig 7). This effect was weak in the coronal plane and strong in the sagittal plane. In addition, in the horizontal plane, the sacrum was located posteriorly. However, the scatter plots of the PC3 score and sacral slope/pelvic inclination showed that these measurements had little correlation (Fig 9). It is possible that the alignment of all pelvises to the anterior pelvic plane at the time of analysis masked the relationship between sacral tilt and pelvic tilt at the original pelvic position. On the other hand, PC3 showed a weak correlation with the Sharp angle.

Reports that young patients in the early stages of DDH have a strong sacral slope [34, 35] are like the characteristics seen in the third principal component. In addition, an increase in sacral slope is closely related to an increase in lumbar lordosis. It is thought that patients with large lumbar lordosis are more likely to develop low back pain caused by the posterior part of the vertebrae [36, 37], and it has been reported that patients with DDH often have low back pain [38]. Thus, the third principal component suggests that it might be related to not only sacral slope, but also the development of low back pain.

The sixth principal component was related to the size of the superior part of pelvis composed of the iliac wings and sacrum (Fig 8). The superior part of the pelvis widens in the Normal group, whereas it hardly widens in the DDH group. Unlike the third principal component, this principal component has strong effects on the coronal plane. Patients with DDH are known to have medial orientation of the iliac wings and anterior inferior iliac spine [14, 39], which is like changes in the sixth principal component. Further, the upper surface of the vertebral body of the first sacral vertebra also tilts anteriorly, suggesting changes in sacral slope. In fact, the scatter plots between PC6 and pelvic measurements showed a weak correlation with iliac wing angle and sacral slope (Fig 9).

The analysis of the present study suggested that the pelvis of DDH has a significant influence not only on acetabular dysplasia, but also on the shape of the acetabulum and sacrum. As shown in the first and sixth principal components, the iliac wings are narrow in the coronal and horizontal planes in the pelvis of DDH. This indicates that the course of the gluteus medius muscle was different from that of the Normal group. It is known that patients with DDH have weak abductor strength [40, 41], but this is likely due to differences in the course of the gluteus medius muscle. In recent years, some researchers reported that gluteus medius muscle weakness is significantly associated with low back pain [42, 43]. This suggests that patients with DDH should also consider gluteus medius training [44, 45] and lower back pain care. Furthermore, a large sacral slope in DDH patients means a large lumbar lordosis. A large lumbar lordosis is also closely related to low back pain [46]. Therefore, this study showed morphometrically that pelvic shape may be associated with gluteus medius muscle weakness and low back pain in patients with DDH.

In summary, the characteristics of the pelvis of DDH patients were identified by comparing the pelvises of 50 patients with DDH and of 50 patients with no obvious deformity using a homologous model. The most important factor in the difference between normal and DDH pelvises was the change in the coxal angle in both the coronal and horizontal planes. That is, in the anterior and superior views, the normal pelvis is a triangle, whereas the DDH was more like a quadrilateral. The pelvic characteristics of DDH patients shown in this study, i.e., changes in the morphology of the iliac wing and sacrum etc., may affect the muscles and lumbar spine to which they are attached and articulated, and treatment that recognizes them should be considered.

## Study limitations

In this study, pelvic morphological change was evaluated in patients with DDH using homologous model analysis, but there were the following problems.

1. Since the pelvic model was simplified, the detailed shape could not be measured.

2. When creating a homologous model, since a perforated (torus-shaped) structure was not allowed, the vertebral foramen and anterior sacral foramen were blocked, resulting in some parts that differed slightly from the actual shape.

3. Pelvic inclination could not be evaluated because all pelvises were aligned in the anterior pelvic plane.

## Supporting information

**S1 File.**
(ZIP)

## Acknowledgments

Dr. Toda Hajime (Sapporo Medical University, Department of Health Sciences) created a conceptual diagram of the homology model in Fig 1. Mr. Toyohisa Tanijiri (Medic Engineering, Kyoto, Japan) developed the software (HBM-Rugle) necessary for homologous model analysis and added several functions according to our requests. He also provided me with the knowledge and information necessary for analysis.

## Author Contributions

**Conceptualization:** Daisuke Suzuki.

**Data curation:** Hiroyuki Takashima, Satoshi Nagoya.

**Formal analysis:** Yui Sasaki.

**Funding acquisition:** Daisuke Suzuki.

**Investigation:** Yui Sasaki.

**Methodology:** Daisuke Suzuki, Ryo Tokita, Hirofumi Matsumura.

**Project administration:** Daisuke Suzuki.

**Resources:** Hiroyuki Takashima, Hirofumi Matsumura.

**Software:** Daisuke Suzuki, Ryo Tokita, Hirofumi Matsumura.

**Supervision:** Hirofumi Matsumura, Satoshi Nagoya.

**Writing – original draft:** Daisuke Suzuki.

**Writing – review & editing:** Daisuke Suzuki, Hirofumi Matsumura, Satoshi Nagoya.

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
