## [Decision Letter · Decision Letter 0]

25 Apr 2024

PONE-D-24-08481Specific pelvic shape in patients with developmental dysplasia of the hip on 3D morphometric homologous model analysisPLOS ONE

Dear Dr. Suzuki,

Thank you for submitting your manuscript to PLOS ONE. After careful consideration, we feel that it has merit but does not fully meet PLOS ONE’s publication criteria as it currently stands. Therefore, we invite you to submit a revised version of the manuscript that addresses the points raised during the review process. A well written article which requires further discussion and diagrams to illustrate the authors findings as well as broader discussion of the literature and whether the conclusions are consistent with the existing beliefs. Similar studies should be discussed in a written format and presented in a tabulated format for readers as well.

We look forward to receiving your revised manuscript.

Kind regards,

Barry Kweh

Academic Editor

PLOS ONE

Journal Requirements:

"Dr. Toda Hajime (Sapporo Medical University, Department of Health Sciences) created a conceptual diagram of the homology model in Figure 1. Mr. Toyohisa Tanijiri (Medic Engineering，Kyoto, Japan) developed the software (HBM-Rugle) necessary for homologous model analysis and added several functions according to our requests. He also provided me with the knowledge and information necessary for analysis.

 This study was supported by JSPS KAKENHI (Grant Number 21K11269)."

"Grant-in-Aid for Scientific Research(C) from Japan Society for the Promotion of Science. Grant Number: 21K11269"

"Grant-in-Aid for Scientific Research(C) from Japan Society for the Promotion of Science. Grant Number: 21K11269"      

Reviewers' comments:

Reviewer's Responses to Questions

**Comments to the Author**

1. Is the manuscript technically sound, and do the data support the conclusions?

Reviewer #1: Yes

Reviewer #2: Partly

2. Has the statistical analysis been performed appropriately and rigorously? 

Reviewer #1: Yes

Reviewer #2: Yes

3. Have the authors made all data underlying the findings in their manuscript fully available?

Reviewer #1: Yes

Reviewer #2: Yes

4. Is the manuscript presented in an intelligible fashion and written in standard English?

Reviewer #1: Yes

Reviewer #2: Yes

5. Review Comments to the Author

Reviewer #1: Thank you for your effort on the study.

The article generally well-written and I think the reader will find it interesting to read.

The introduction section please state the aim of your study.

The methods and the results section is well-written but please add comparative data like Sharp angle, sacral slope and other measurements between normal and DDH group.

In discussion section, at first paragraph please remove "Although morphological analysis using a homologous model assumes that all vertices comprising the pelvis are homologous, it is an objective method that uses all morphological information. This method has recently been used to evaluate foot form, cranial morphology, facial aging changes, and pelvis sex determination, and it is attracting attention as a new comparison method for 3D models [15, 19, 21, 24]" and move to next or further paragraph.

In the first paragraph clearly state the most important findings of your study.

Reviewer #2: I would like to thank the authors for their efforts and valuable contributions.

In the title of the article, it is stated that specific pelvic shapes in patients with developmental hip dysplasia were examined by model analysis. However, it is stated that all DDH patients used in the modeling underwent rotational osteotomy surgery. In fact, these patients should be stated as DDH patients who underwent surgery. The age range of this patient group is given as 21-52 years. It should be stated at what age the patients had surgery and how long after the surgery the modeling was done.

6. PLOS authors have the option to publish the peer review history of their article (what does this mean?). If published, this will include your full peer review and any attached files.

Reviewer #1: No

Reviewer #2: **Yes: **Dr. Murat Çakar

---

## [Author Response · Author response to Decision Letter 0]

16 May 2024

The authors would like to thank the reviewers and editors for their time and effort to review this manuscript. 

Reviewer #1

The introduction section please state the aim of your study.

The aim of our study added to introduction.

The methods and the results section is well-written but please add comparative data like Sharp angle, sacral slope and other measurements between normal and DDH group.

Measurement values such as Sharp angle etc. are shown in the table 2.

In discussion section, at first paragraph please remove "Although morphological analysis using a homologous model assumes that all vertices comprising the pelvis are homologous, it is an objective method that uses all morphological information. This method has recently been used to evaluate foot form, cranial morphology, facial aging changes, and pelvis sex determination, and it is attracting attention as a new comparison method for 3D models [15, 19, 21, 24]" and move to next or further paragraph.

In the first paragraph clearly state the most important findings of your study.

The sentences you pointed out were moved to the second paragraph.

Reviewer #2: 

In the title of the article, it is stated that specific pelvic shapes in patients with developmental hip dysplasia were examined by model analysis. However, it is stated that all DDH patients used in the modeling underwent rotational osteotomy surgery. In fact, these patients should be stated as DDH patients who underwent surgery. The age range of this patient group is given as 21-52 years. It should be stated at what age the patients had surgery and how long after the surgery the modeling was done.

The 3D pelvic models of a DDH patients were reconstructed from CT images before rotational acetabular osteotomy. We added this information on page 5, lines 88-90.

---

## [Editor Report · Decision Letter 1]

21 May 2024

Specific pelvic shape in patients with developmental dysplasia of the hip on 3D morphometric homologous model analysis

PONE-D-24-08481R1

Dear Dr. Suzuki,

We’re pleased to inform you that your manuscript has been judged scientifically suitable for publication and will be formally accepted for publication once it meets all outstanding technical requirements.

Kind regards,

Barry Kweh

Academic Editor

PLOS ONE

Additional Editor Comments (optional):

The authors have submitted a well written manuscript and addressed the reviewers comments.

---

## [Editor Report · Acceptance letter]

24 May 2024

PONE-D-24-08481R1 

PLOS ONE

Dear Dr. Suzuki, 

I'm pleased to inform you that your manuscript has been deemed suitable for publication in PLOS ONE. Congratulations! Your manuscript is now being handed over to our production team.

Kind regards, 

on behalf of

Dr. Barry Kweh 

Academic Editor

PLOS ONE